# Fast in-line failure analysis of sub-micron-sized cracks in 3D interconnect technologies utilizing acoustic interferometry
Priya Paulachan [1], René Hammer[1], Joerg Siegert[2], Ingo Wiesler[3] & Roland Brunner [1] ✉

More than Moore technology is driving semiconductor devices towards higher complexity and further miniaturization. Device miniaturization strongly impacts failure analysis (FA), since it triggers the need for non-destructive approaches with high resolution in combination with cost and time efficient execution. Conventional scanning acoustic microscopy (SAM) is an indispensable tool for failure analysis in the semiconductor industry, however resolution and penetration capabilities are strongly limited by the transducer frequency. In this work, we conduct an acoustic interferometry approach, based on a SAM-setup utilizing 100 MHz lenses and enabling not only sufficient penetration depth but also high resolution for efficient in-line FA of Through Silicon Vias (TSVs). Accompanied elastodynamic finite integration technique-based simulations, provide an in-depth understanding concerning the acoustic wave excitation and propagation. We show that the controlled excitation of surface acoustic waves extends the contingency towards the detection of nm-sized cracks, an essential accomplishment for modern FA of 3D-integration technologies.

Three-dimensional integration technology is a continuously burgeoning field in the semiconductor industry that addresses the convergence of Moore's law and More than Moore (MtM)[1]. The third dimension helps to configure highly integrated heterogeneous devices by vertically stacking different functional components together with a notably reduced form factor[2]. Through Silicon Vias (TSVs) represent a high-performance 3D integration technology used in microchip engineering to build sophisticated and multifunctional microelectronic setups like microelectromechanical systems (MEMS), inductors, power electronics, optical sensors, and 3D packages[3–5] with low cost and miniature size[5,6]. TSVs are fabricated through an assortment of various steps including the etching of deep trenches through the silicon substrate, followed by the deposition of electric insulation layers (e.g. silicon dioxide). Thenceforth the TSV is either filled or coated with a conductive metal, leading to filled or open TSV technology respectively[5,7–9]. One of the key factors determining the abiding reliability of TSVs is the thermo-mechanical stress resulting from the mismatch of the coefficient of thermal expansion (CTE) between the TSV conductive path and the bulk[10,11]. Tungsten-lined open TSV technology addresses this issue of CTE mismatch but at the cost of high residual stresses arising after the deposition process within the thin film stack formed at the sidewall and bottom[12,13]. In general, the formation of residual stresses in thin films is a challenging problem[14–16]. The emerging residual stress may cause critical damage formation such as delamination and/or cracks in the sidewall or bottom of the TSV. Therefore, defect detection in TSVs on μm-level and below is crucial to ensure a reliable fabrication of interconnects in the production[17]. However, failure analysis of TSVs in-line within an industrial environment poses a tremendous challenge. In particular, the requirement such as the non-destructive high-resolution characterization of defects with micron- and sub-micron size in a time and cost-efficient manner, as well as to gain a statistical relevant amount of TSVs on wafer level in-line, is tough to fulfill[18].

In the past several decades, there have been several advancements in the field of quality assessment of microelectronic components[19–25]. This includes methods such as automatic optical inspection (AOI)[26,27], scanning electron microscopy (SEM)[20,28], emission microscopy (EMMI)[21], micro-X-ray computed tomography (μ-XCT)[29–31], X-ray microscopy (XRM)[32–34], terahertz (THz) techniques[32], transmission electron microscopy (TEM)[35], electrical characterization techniques[36], scanning acoustic microscope (SAM)[37] and many more[38,39]. Choosing the most suitable testing method among these techniques is crucial when it comes to perform a thorough

[1]Materials Center Leoben (MCL) Forschung GmbH, Leoben, Austria. [2]ams-OSRAM AG, Premstaetten, Austria. [3]PVA TePla Analytical Systems GmbH, Westhausen, Germany. ✉e-mail: roland.brunner@mcl.at

non-destructive analysis of defects in high throughput data, such as those found in 3D interconnect technologies. Certain positive attributes of these methods qualify them to meet some of the above-stated requirements for assuring high yield and high quality fabrication of TSVs, but not all of them are met[40]. AOI is a non-destructive technique[26], which excels in identifying defects on the bottom surface of TSVs but falls short when it comes to detecting sidewall defects[17]. EMMI is an optical defect inspection technique to detect and localize failures in integrated circuits[21]. EMMI is primarily effective on electrically active structures and defects with electrical signatures, hindering its application to certain critical defects. SEM offers high-resolution images for both sidewall and bottom TSV areas. However, it faces limitations in high-throughput inspection and statistical information generation due to time-consuming data acquisition[20,28]. The X-ray based imaging techniques[41] such as µ-XCT or XRM offer distinct advantages but they also come with drawbacks that constrain their applicability for defect detection in TSVs. µ-XCT represents a well-established non-destructive, 3D imaging technique employed for the analysis of defects within the micrometer range in 3D semiconductor packaging[31]. Here the possible resolution is defined by the focal spot size, which is about a micron. Further the magnification and resolution is dependent on the sample size[42] due to the utilized cone beam configuration. The lack of resolution in the nm-range and the extended duration time for µ-XCT measurements poses a major challenge[31], constraining the generation of comprehensive statistical information for sub-micron crack detection. Those constraints could potentially affect the overall reliability of quality assessments of TSVs at wafer level. XRM is proficient in identifying voids, delamination, or cracks in three dimensions[32]. It utilizes additional optics, which makes the magnification or resolution less dependent on the sample size. Nevertheless, there are certain limitations with respect to resolution and contrast[32]. Further the time required for imaging the entire package is a downside[32]. THz techniques have the capability to detect delamination, voids, and cracks, however, the assessment of defect presence requires the availability of reference data. The method also shows low resolution compared to XRM[32]. Electrical characterization techniques helps in identifying and rectifying deviations in electrical properties that might affect the reliability of semiconductor devices, but provides limited insights into the structural integrity of the device[36].

Among these mentioned methods, SAM is notable for its exceptional characteristics of the fast, cost-efficient and non-destructive capability of damage analysis over a large array of 3D interconnects[40,43]. Nevertheless, the achievable image resolution using SAM for failure analysis of 3D interconnects is of high intrigue[18]. The resolution capabilities of a SAM are strongly governed by the wavelength of the acoustic waves and thus the frequency of the transducer[44]. There have been several studies carried out on defect detection in TSVs using SAM within different frequency ranges[18,45–49]. The improvement in lateral and axial resolution is possible with high-frequency transducers, but results in limiting analysis capabilities to defects close to the surface[50]. The resolution limit of MHz frequency transducers is thus driving the industrial application towards the GHz regime. For example, in the analysis of voids in materials used in micro-connects such as the Cu-Sn material system[51], for the assessment of the adhesive bonding in semiconductor wafers[52], or the identification in poor layer-to-layer adhesion in integrated circuits (ASIC)[53] and so forth. The high frequency transducers are not ideal for quality assessment in open TSV technology, because of the intricate analysis required for the TSV bottom. This demands a modified SAM setup that can extract information about the TSV sidewall as well as TSV bottom, unlike GHz SAM setup.

Interferometry-based characterization approaches have a wide range of applications in various fields of science and engineering[54–56]. There are several interferometry techniques based on Moiré interferometry (MI)[57], holographic interferometry (HI)[58], optical interferometric based techniques like speckle interferometry (SI) and electronic speckle pattern interferometry (ESPI)[59]. Those interferometry-based techniques provide advantages with respect to failure assessment e.g. deformation classification in microelectronic packages[60], subsurface defect inspection on silicon

wafers[59], or solder joint failure investigations[60]. In acoustics, constructive and destructive interferences may occur between elastic waves and surface acoustic waves (SAWs)[44]. The alluring feature of SAWs is their energy confinement at the near surface of solids[61,62]. SAWs are therefore capable in modifying the direct reflections from the substrate to provide an interference effect. This property of SAWs is utilized in various applications ranging from measuring intrinsic material properties to electronic devices[61–63], geophysics, material science, telecommunications, photonics and many more[64].

In this paper, we apply the concept of acoustic interferometry for cost and time efficient high-resolution failure analysis in TSVs down to the nanometer regime by amalgamating special transducer configurations. The transducer is suitable to excite surface acoustic waves with a modified but highly industrial serviceable scanning acoustic microscope setup. In particular, we use customized transducers with a central frequency of 100 MHz and lenses with wider opening angle, exceeding the critical Rayleigh angle and enabling the controlled excitation of Rayleigh waves or SAWs. We conduct, accompanied to the experimental results, wave propagation simulations based on elastodynamic finite integration technique (EFIT). The simulations aim to validate the generation of SAWs utilizing the SAM interferometry setup on wafer level as well as to contribute to a deeper understanding of SAW propagation, with a specific focus on the angles of the acoustic lenses. The detected different SAM interference patterns can be associated in an automated manner to either non-defective or various defective TSV classes, by utilizing an end-to-end convolutional neural network[17]. The presented methodology shows that the controlled excitation of surface acoustic waves facilitates the detection of nm-sized cracks, an essential accomplishment for modern failure analysis in 3D integration technologies. We point out that the discussed acoustic interference approach is not only limited to failure analysis of TSVs but can be also exploited to further topics.

## Results and discussion
### Acoustic interferometry setup
Figure 1a illustrates the workflow for the SAM interferometry setup. We conduct a narrow band, tone burst driven setup as illustrated in Fig. 1a suitable to improve the acoustic wave detection capabilities and to deliver failure analysis possibilities below the µm regime. The presented SAM interferometry approach involves a strategic choice in balancing the penetration depth and resolution. In general, the frequency of the acoustic wave directly relates to the resolution, see method section. Increasing the frequency of the acoustic wave results in higher resolution, thus smaller structures can be resolved. However, also the attenuation in the coupling liquid and in the sample increases with frequency. For this reason, higher frequency leads to a lower penetration depth. Therefore, in many cases, a trade-off between resolution and penetration depths has to be made for conventional SAM measurements. Notably, in the SAM interferometry approach, the detection capability does not solely depend on the frequency of the transducer. Rather the excitation of additional wave modes such as SAWs by utilizing a modified SAM setup and lenses with a wider opening angle (60°) facilitate the detection of defects smaller than the resolution commonly determined by the transducer frequency. The interaction of these excited SAWs with other wave modes within the TSV henceforth are mapped in the measured SAM C-scan images leading to characteristic interference patterns. The proposed approach of SAM interferometry is contrary to the usual approach and less dependent on the transducer frequency. It exceeds existing resolution capabilities but also provides the possibility to enable sufficient penetration depth.

The modified SAM setup, as shown in Fig. 1a, aims for the controlled generation of interference effects based on the superposition of surface acoustic waves and bulk acoustic waves. We use spherical lenses with a specific lens aperture and piezoelectric transducer. An accurate positioning system is applied to move the transducer from the focused to different defocused positions in z-direction, see Methods. The precise movement of the x-y scanning stage enables the controlled excitation of surface acoustic

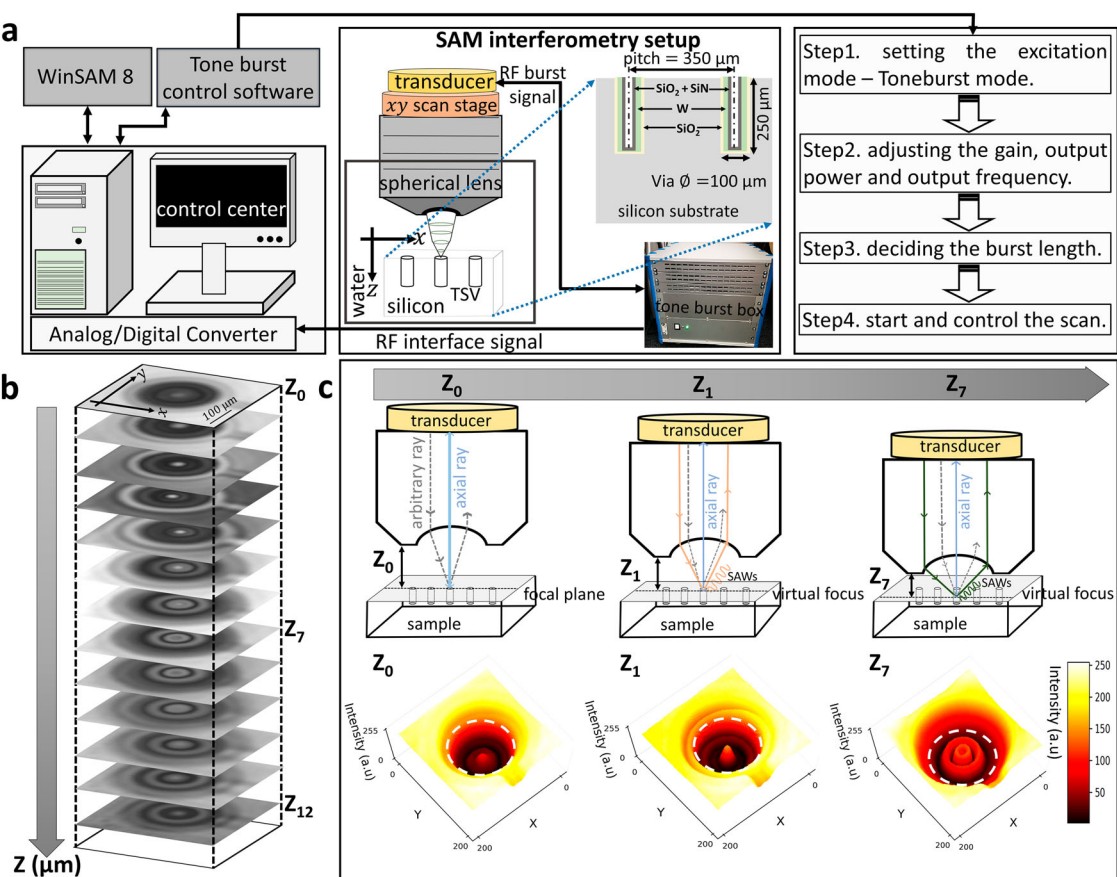

**Fig. 1 | Experimental scanning acoustic microscope interferometry setup for TSV characterization. a** Overview of the modified SAM interferometry setup with incorporated transducer, signal processing and control for efficient failure analysis of TSVs. The four steps are indicated on the right to perform the acoustic interferometry. A schematic of the investigated TSV structure is further illustrated. **b** C-scan images obtained at different focusing heights ($Z_0$,…, $Z_7$,…, $Z_{12}$). $Z_0$ is the position at which the transducer is focused on to the surface of the sample ($Z_0 \gg Z_{12}$). The step size between each Z positions is 20 µm. **c** The schematic ray diagram

illustrates the possible excitation of distinct wave modes for different Z position e.g. $Z_0$, $Z_1$ and $Z_7$. At $Z_1$, the axial rays and the rays exciting surface acoustic waves or Rayleigh waves can generate interference effects. Corresponding to each respective Z position, the accompanying intensity plots in the x-y plane is shown to illustrate the formation of the fringes due to interference effects. The position of the TSV is indicated by the white dashed line. Intensity plot with low intensity (black), to high intensity (yellow) in the x-y-plane.

waves at any position below the focused position $Z_0$. The SAM interferometry set up is operated in pulse-echo mode. That means, the same transducer is capable in transmitting and receiving the acoustic waves. As shown in Fig. 1a, the transducer is wired with a custom-engineered tone burst box setup, which provides the possibility to perform high precision frequency dependent analysis from the reflected transducer signal. Further a schematic of the investigated TSV structure including the relevant dimensions is provided.

Four adjustment steps are necessary to generate a tone burst signal that is useful for signal analysis. The first step is to set the excitation mode to tone burst mode. The second step is to set the frequency of the tone burst signal. The selection of the tone burst frequency depends on the information that needs to be extracted from the sample under investigation. The careful adjustment of the frequency makes our SAM interferometry approach more efficient and highly suitable for high-resolution defect analysis. The third step is to determine the duration of the burst. The duration of the burst has to be set in accordance with the wavelength of the acoustic wave to achieve the best possible spatial resolution. The final step is dealing with the start and control of the scan, using the designed burst signal. The piezo electric transducer converts the RF electrical signal to acoustic waves, which then propagate towards the sample through the acoustic lens. The opening angle of the lens is chosen to reach an angle of incidence above the Rayleigh angle of silicon, enabling the excitation of bulk as well as surface acoustic waves. The reflections from the sample are converted to a digital signal using an

analog to digital converter (ADC) and can be visualized in the form of C-scan images using the WinSAM software.

In particular, the utilization of the tone burst signal with a suitable frequency and pulse duration is critical for improving the detection of defects in TSVs with varying geometries. TSVs, being intricate three-dimensional structures, require a method that can adapt to their specific dimensions and potential flaws. The short burst duration and carefully chosen frequency of a tone burst signal enhances the resolution of defect detection. This selection allows the precise identification of smaller defects within the confined space of TSVs. TSVs with smaller dimensions require a tone burst signal with higher frequencies for effective detection, while TSVs with larger dimensions benefit from lower frequencies to ensure sufficient penetration of the acoustic waves into the sample. The length of the tone burst signal, or pulse duration, influences the temporal resolution. The tone burst signal's ability to provide enhanced contrast proves invaluable for distinguishing subtle variations in acoustic impedance, aiding in the clear differentiation between TSV materials and potential defects. Moreover, the short duration of the signal minimizes interference and crosstalk, particularly crucial when TSVs are closely located. The compatibility of the tone burst signal with the acoustic lenses further refines SAM's imaging capabilities, ensuring high-resolution examination of the complex TSV structure. For the utilized transducer frequency of 100 MHz, a tone burst signal with 50 ns duration is selected, since it provides better isolation of the signals. Further details in this context are provided in the Method section.

Figure 1b shows the C-scan for different transducer positions along the z-direction from $Z_0$ to $Z_{12}$. The schematic shown in Fig. 1c illustrates the possible ray propagation exemplarily for the transducer position $Z_0$, $Z_1$, and $Z_7$. Here we consider mainly normal incidence. At the focus position indicated with $Z_0$, mainly the axial rays propagate from the sample along the axis of the lens. Rays from the lens which are specular reflected by the sample, shown as arbitrary rays in Fig. 1c, do not necessarily contribute to the excitation of the transducer[44]. Once the transducer is defocused with the condition $Z_n < Z_0$, $n = 1, 2, \ldots$, then in addition to the rays reflected along the axis of the lens, further rays can be excited that are incident on the sample at the Rayleigh angle[44]. In case of the defocused transducer, both axial rays and the rays exciting surface acoustic waves or Rayleigh waves may then contribute to the transducer signal. Therefore, at the transducer the complex-valued sum of the axial ray and additional generated rays can be detected. Interference effects between those rays become then apparent[44]. Once a defocused position of the transducer is reached ($Z < Z_0$), concentric intensity fringes unlike seen in the focused position emerge near the intensity associated with the TSV circumference within the C-scan image data. Those fringes change in intensity as the position in Z is altered as shown in Fig. 1c. While Rayleigh waves can be excited at various positions below $Z_0$, the selection of the optimal defocus position is crucial for achieving the best possible contrast and effectively revealing any underlying inhomogeneity. The optimum position of Z suitable for our experimental SAM setup, with respect to the intensity, Rayleigh velocity and formed fringes, is sustained at $Z_7$ with a Z of $-140\,\mu m$, revealing a Rayleigh velocity of 5200 meters per second, closely aligning with the theoretical value[65], see Supplementary Note 1. At defocused position $Z_7$, a well-defined intensity pattern with evenly spaced fringes of width ($\lambda/2$) can be seen[66] unlike $Z_1$, which in turn indicates the stronger interaction of the Rayleigh wave.

Further information about the selection of the Z-position is provided in Supplementary Note 1 and Supplementary Fig. 1. The results prove qualitatively that the excitation of surface acoustic waves at any position below the focus position takes place as well as indicates the position in Z maximizing the interference effect.

## Acoustic interferometry for high-resolution sub-micron crack detection

Figure 2a illustrates the SAM C-scan image of a TSV test array at the defocused position $Z_7$, maximizing the interference effect. Here, we efficiently evaluate the quality of 308 TSVs using the SAM interferometry setup in approximately 15 minutes at 2 μm/pixel resolution with a 100 MHz transducer and a lens providing an opening angle of 60°. A frequency of 100 MHz is selected to provide a sufficient balance between resolution and penetration depth. The achieved fast end time, sets the developed approach apart from other FA techniques like SEM, XCT etc. Therefore, the setup is highly suited for the inline inspection of TSVs within an industrial environment. We utilize the convolutional neural network methodology developed in ref. 17 for the automated and time-efficient localization and classification of the TSVs on wafer level. This workflow involves two sequentially connected convolutional neural network (CNN) architectures. The first CNN (CNN1) detects TSV locations using a 100 × 100 pixels sliding window approach. CNN2 classifies thus detected TSVs based on their quality. CNN1 is trained with labeled images, featuring two classes: TSVs centered in the bounding box and TSVs off-center or C-scan images with the background. During testing, CNN1 predicts multiple bounding boxes, refined by non-maximal suppression (NMS) to select the best bounding box, thus identifying TSVs as objects. CNN2, taking CNN1 predictions as inputs, is trained with C-scan images showing blossom-shaped interference patterns, in addition to classes defined in the previous study[17]. For the training of CNN1, we use a dataset consisting of a total of 20,000 images. We split it into 70% (14,000 images) for training and 30% (6000 images) for testing. For CNN2, a dataset of 10,000 images, each sized 100 × 100 pixels, showing interference patterns of different TSVs is used. Out of these, 7000 images are used to train CNN2, and 3000 are kept aside to validate the model. Since the blossom-shaped patterns are less common

compared to other classes of interference patterns, we used augmentation techniques like flipping, rotation, and random zoom-in and -out on the C-scan images to make the dataset larger and more diverse. Figure 2b depicts an exemplary region of interest (ROI) and illustrates the automated classification and localization of the TSV array using E2E-CNN in ref. 17. Here, within the ROI we extract three classes indicated by different color codes. As shown in Fig. 2c, *Class 1* indicates homogeneous interference fringes. *Class 2* illustrates patterns with a deterioration of the homogenous fringes. Both classes are similar to what has been shown in refs. 17,18,49 where such patterns could be associated with non-defected TSVs and defected TSVs, respectively. Correlated SEM investigations disclosed failure sizes within the μm-regime. Beyond the state of the art, we reveal an additional class illustrating a further unique interference pattern. The observed pattern is indicated as *Class 3* in Fig. 2c. The interference pattern for *Class 3* is notably different to *Class 1* and *Class 2* or to other classes reported previously[17,40,49]. Figure 2d shows the corresponding intensity plot of *Class 3* in the x-y plane for the C-scan image, which indicates the disturbance of the interference patterns within the TSV along the whole circumference.

A distinction between different interference pattern classes manifested in the C-scan is not feasible with a conventional SAM. In Supplementary Fig. 2, we illustrate a comparison between the interferometric C-scan image and the conventional SAM C-scan image for two TSVs. The conventional C-scan image shows for TSV1 and TSV2 similar results. In case of the interferometric C-scan image, a clear difference between TSV1 and TSV2 is depicted. TSV1 and TSV2 show two different interference patterns which correspond to *Class 3* and *Class 1*, respectively. The interference pattern associated with *Class 1* and *Class 3* in Fig. 2c as well as **Fig. S2**, are generated by the interaction of SAWs with the TSV side wall. In case of *Class 3*, the SAWs are interacting with a sub-micron crack located in the sidewall of the TSV, leading to the disturbance of the interference patterns. Correlative scanning electron microscopy characterization of a TSV, associated with a *Class 3* interference pattern, shows a thin crack extending over the whole circumference of the metallized side wall, see Fig. 2e. In Supplementary Fig. 3 we show further TSVs for *Class 3*. Again, the associated SEM images illustrate for those investigated TSVs a thin nm-sized crack within the sidewall. Figure 2f shows the cross section, performed via ion slicing, of the TSV associated with a *Class 3* interference pattern. Notably, a cone crack in the metallization with a crack opening of about 200 nm, much smaller than the lateral and axial resolution of the 100 MHz transducer, is featured. Hence, the acoustic interferometry is suitable to detect features sizes below the resolution limit of the transducer frequency due to the excited SAWs. The SAM-based failure analysis is carried out without any prior knowledge of the defect position.

## Excitation of surface acoustic waves within the TSV

We conduct numerical simulations based on the elastodynamic finite integration technique (EFIT)[67,68] to gain an improved understanding of the excitation of wave modes at different transducer positions relative to the TSV. The primary objectives of the EFIT simulations are threefold. Firstly, the simulations aim to validate the generation of SAWs utilizing the SAM interferometry setup on waver level, setting it apart from conventional SAM methods. Secondly, the EFIT simulations aim to contribute to a deeper understanding of SAW propagation, with a specific focus on the angles of the acoustic lenses. Moreover, the goal includes the utilization of simulation outcomes in refining the acoustic lens for optimization and the development of a more advanced lens design.

The flow chart of the entire process of EFIT simulation is shown in Fig. 3a. Further details are provided in Supplementary Note 2. We assume an isotropic linear elastic medium to uncover especially the role of the SAWs in context to the generated interference fringes. The utilization of the EFIT helps to understand the propagation and excitation of acoustic waves in liquids and solids. The simulation workflow is outlined through four key steps: initialization, solver definition, time stepping, and post-processing. The initial stage encompasses the setup of the simulation domain, including the definition of geometries such as the acoustic lens, transducer, and

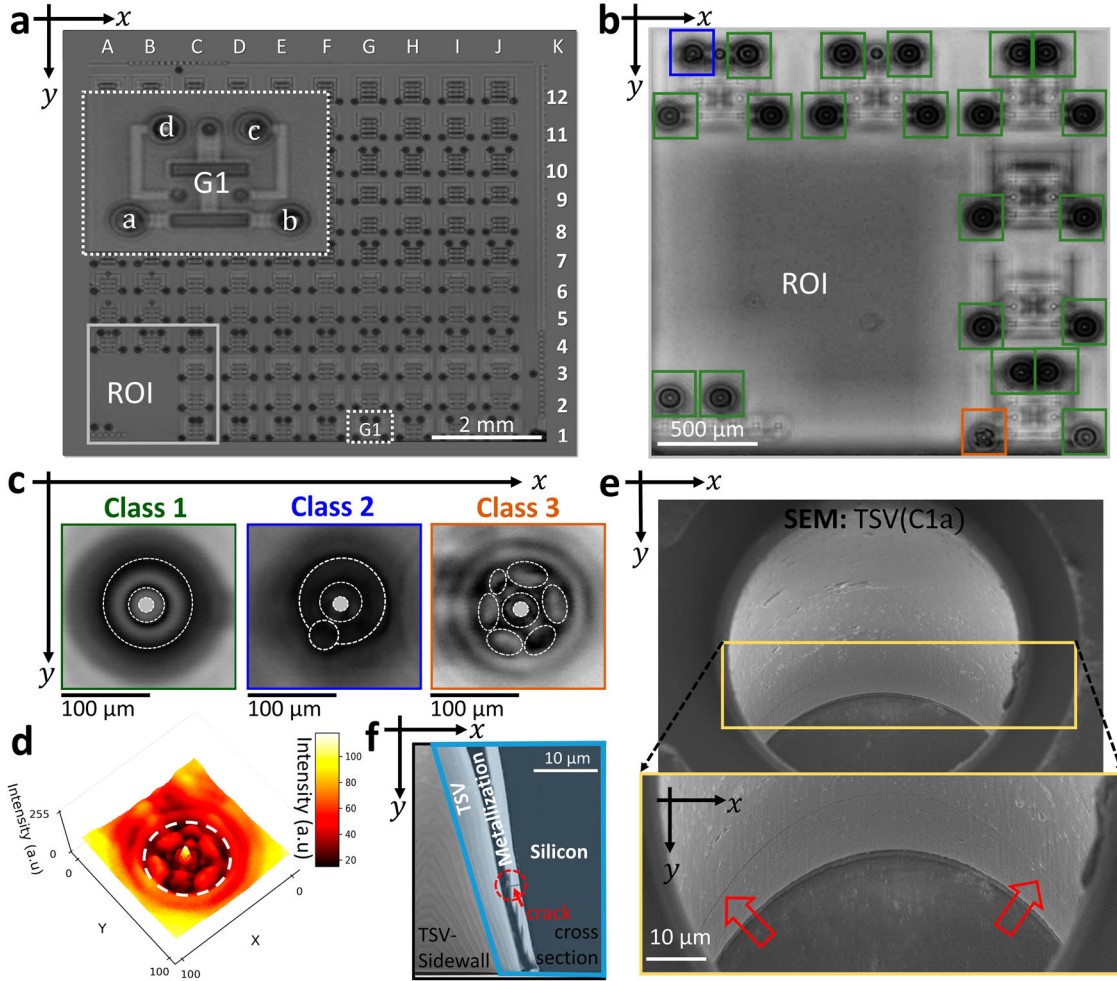

**Fig. 2 | Non-destructive detection of defects in TSVs with high resolution utilizing acoustic interferometry. a** SAM C-scan image showing a TSV array with a subgroup of array marked as G1. The letters a, b, c and d denote the position of TSVs within the subgroup G1. **b** Automated classification and localization of the TSVs within the exemplary ROI utilizing the End-to-End convolutional neural network (E2E-CNN). TSVs with homogeneous fringes are predicted as *Class 1* (green) TSVs. *Class 2* (blue) indicates a single inhomogeneity. *Class 3* (brown) shows a blossom-like shaped interference pattern. The predictions of E2E-CNN are color coded in green, blue and brown for *Class 1*, *2* and *3*, respectively. **c** Magnified SAM C-scan images of the TSVs associated with *Class 1*, *Class 2* and *Class 3*, highlighted with green, blue and brown frames, respectively. **d** Intensity plot (low intensity corresponds to black, yellow to high intensity) in the x-y-plane for *Class 3* at the defocused position $Z_7$. The distorted fringes along the circumference can be seen leading to a blossom-shaped pattern. White dashed line indicates the position of the TSV. **e** SEM images illustrating 'TSV(C1a)' and indicating a crack, with an opening of about 200 nm, within the metallized wall of the TSV. The crack is better resolved in the magnified image (yellow box). Two red arrows are used to highlight the crack position. **f** Cross sectional SEM image (after ion slicing) of the crack in 'TSV(C1a)' with a cone crack. The crack is indicated within the red dotted circle and pointed using a red arrow marker.

sample, along with their corresponding material parameters. The transducer's excitation is modeled as a modulated sinusoidal wave with a Gaussian envelope, emulating a tone burst signal. The frequency and length of this tone burst signal is set in accordance with the SAM interferometry setup. The input stage also involves determining stable spatial and time step sizes. The spatial step size is determined by the minimum wavelength of shear waves in the medium. For our simulations, 10 grid points per wavelength are used as recommended in literatures[69]. Additionally, the Courant–Friedrichs–Levy (CFL) condition is applied to calculate the upper limit of the time step size, see Method section. An absorbing boundary condition is also defined at this stage to attenuate the reflections at the domain boundaries. The second step of the simulation focuses on grid generation based on the calculated spatial step size, followed by defining the necessary equations of EFIT. Finite difference equations for stress and particle velocity are derived from integral forms of Cauchy's equation of motion and Hooke's law. As the simulation progresses in time during the third step, the velocity fields are updated, followed by the updating of stress tensors using the previously updated velocity components. Here, the velocity and stress variables are discretised on a staggered spatial and temporal

grid[70]. This discretization uses the central difference operator. The simulation proceeds in time within a leap-frogging scheme[71]. Supplementary Fig. 4 shows the discretization of variables on a unit cell with the necessary equations for the simulation of the elastic waves, see also Supplementary Note 2 for further information. The final stage of the simulation involves the post-processing step, as illustrated in the flow chart.

For an improved understanding of the SAW propagation in the SAM interferometry setup, we transfer the exact transducer position in Z (focus and defocus) to the EFIT simulations, see further details in Supplementary Fig. 5 and Supplementary Note 3. The setup used in the EFIT simulations consists of an acoustic lens with an anti-reflective coating (ARC) in compliance with the experimental setup. The sample within the simulation is defined by the silicon substrate and a TSV with a diameter of 100 μm and a depth of 250 μm configured by different layers to mimic the metallization present in the real sample, see Supplementary Fig. 6. The applied parameters for the simulation such as the Young's moduli, Poisson ratios and mass densities of silicon, various layers of TSV such as silicon dioxide, tungsten, silicon nitride and sapphire for lenses used in defining various materials are summarized in Table 1.

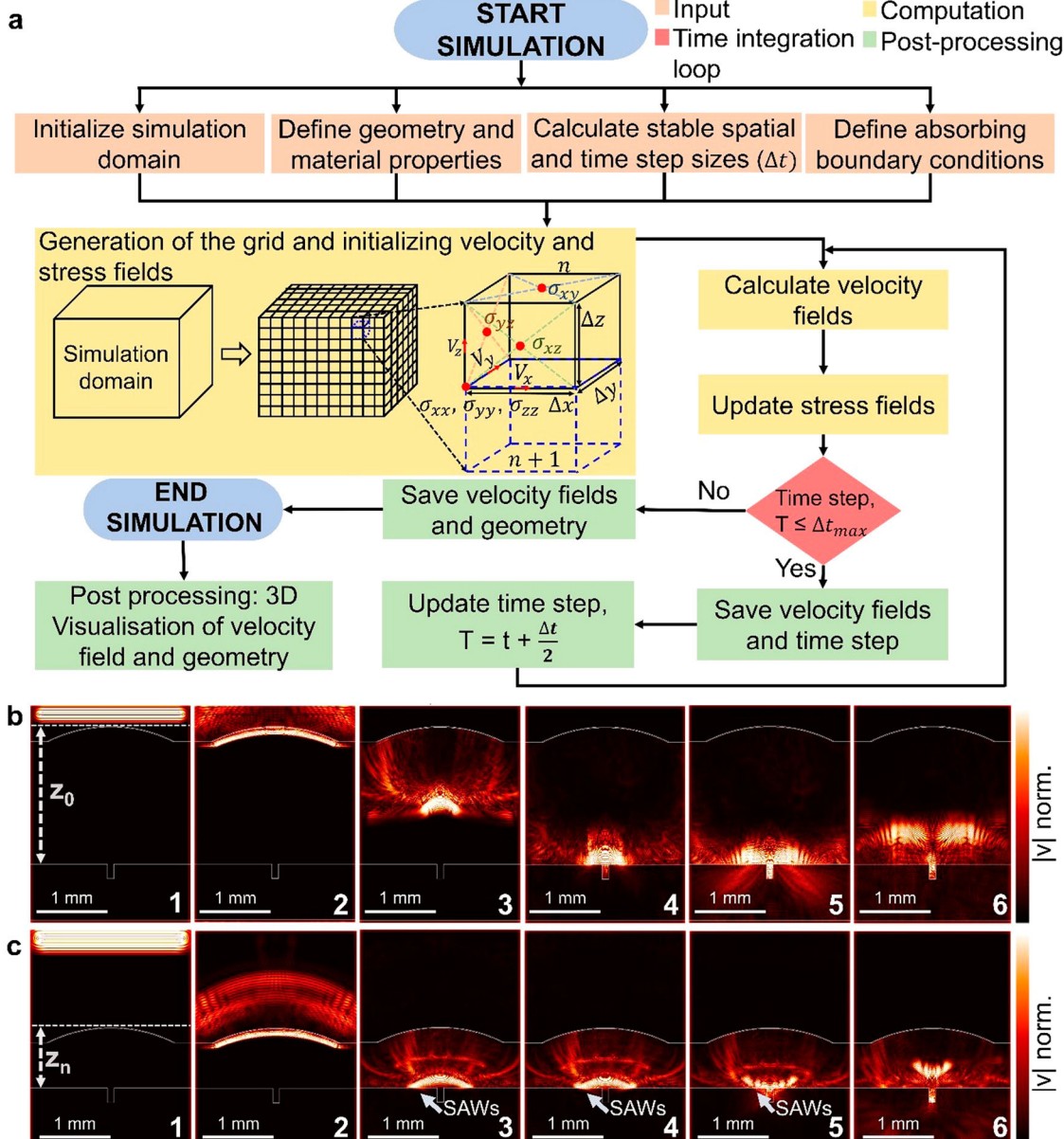

**Fig. 3 | Simulated wave propagation at focused ($Z_0$) and defocused position ($Z_n$) using 100 MHz, 60° transducer for different time steps. a** Flow chart of the EFIT simulation. **b** EFIT simulation at the focused transducer position. **c** EFIT simulation at the defocused transducer position. The excitation and propagation of surface acoustic waves (SAW)s - see time steps 3 and 4 - are shown. The arrow marker in time step 3-4 highlights the generated SAWs at the surface. Time step 5 and 6 show the interaction of the SAWs with the TSV. The latter renders the possibility to detect inhomogeneities within the TSV based on the detected interference patterns. Intensity for the velocity ranges from yellow (high) to black (low).

**Table 1 | Summary of material parameters for 2D and 3D elastodynamic finite integration technique (EFIT) simulations[44,74–76]**

| Material | Young's Modulus (GPa) | Poisson's ratio | Mass density (kg m$^{-3}$) |
|---|---|---|---|
| Silicon | 130 | 0.28 | 2330 |
| SiO$_2$ | 70 | 0.19 | 2200 |
| W | 405 | 0.28 | 18250 |
| SiN | 140 | 0.27 | 3185 |
| Sapphire | 345 | 0.29 | 3980 |

Figure 3b, c shows the full lens simulation with a slice representation utilizing the EFIT simulation with the transducer positioned at the focus and defocus position, respectively. Within time step 1, an acoustic wave at the center frequency of 100 MHz is excited, see time step 1 in Fig. 3b and c, respectively. Here, a plane wave propagates towards the ARC within the defined acoustic lens. The velocity of the propagating plane wave is approximately 11,100 meters per second. The high difference of sound velocity of the sapphire lens compared to the liquid coupling medium causes the acoustic wave to be refracted into a spherical wave front[72], see time step 2 in Fig. 3b and c, respectively and further propagates towards the sample in time step 3. At the focused position $Z_0$ of the transducer, the propagating wave front converges near the center of curvature of the acoustic lens, see time step 4 in Fig. 3b. The simulation at time step 4 and 5 in Fig. 3b exhibits that one fraction of the acoustic wave is reflected by the silicon, i.e. propagating back to the lens, while the other fraction is propagating towards the

bottom of the TSV. Finally, in time step 6 of Fig. 3b the propagating acoustic wave is reflected from the bottom of the TSV.

Differently the situation for the defocused transducer in time step 3 and 4, as shown in Fig. 3c. Here, an additional wave mode near the surface of the TSV is excited. The excitation of this wave and its interaction with the TSV is marked in Fig. 3c. Such additional wave modes can be also excited in filled TSVs as shown in the EFIT simulation in Supplementary Note 4 and Supplementary Fig. 7. Such excited waves, which can be linked to the surface acoustic waves, are manifested within the observed interference fringes of the C-scan data, see Supplementary Note 1 and Supplementary Fig. 1. The evaluated depth information from the sample at a single X-Y location (gated A-scan signal) depicts the corresponding gray value in the SAM C-scan images. We validate the simulation result of Fig. 3c by comparing the simulated A-scan signal at the defocused position ($Z_n$) with the measured SAM A-scan signal, see Supplementary Fig. 8. The simulated A-scan signal shows a high resemblance to the experimental A-scan signal, which in turn validates the simulation. For the excitation of SAWs, also the position of the acoustic lens with respect to the x-y plane of TSV needs to be considered, see Supplementary Note 5 and Supplementary Fig. 9. The simulation result in Supplementary Fig. 9 shows that the interaction of SAWs depends on the lens positioning relative to the TSV irrespective of the pitch length between the adjacent TSVs.

### Acoustic lens selection and impact on failure analysis

The excitation of the surface acoustic waves is dependent on the transducer position with respect to the sample. Here, an optimized position at $Z_7$ with $-140\,\mu m$ is achieved leading to a Rayleigh velocity of 5200 meters per second which is in accordance with the velocity of Si[65]. In order to investigate the impact from the acoustic lens angle on the SAW generation and defect detectability, we perform 3D EFIT simulations with a 100 MHz transducer as used in the experiment. Here, we keep the position of the lens in Z constant. The result for an opening angle of 60° and 80° is shown in Fig. 4a and b, respectively. The larger curvature of the 80° lens results for the same defocus position in a smaller working distance compared to the 60° lens. Further details with respect to the 3D EFIT simulation are presented in the method section.

In Fig. 4a, b the simulation step 1 starts with the generation of the acoustic waves at the silicon surface. The complete series of 3D EFIT simulation results, using a 100 MHz 60° and 80° lens configuration, is shown in the Supplementary Movies 1 and 2. The energy of the wave in time step 1 is concentrated near the surface for both opening angles. The simulations at time steps 2 for 60° and 80° manifest the propagation of SAWs towards the TSV. Time step 3 in Fig. 4a, b indicates the position in z-direction where the generated SAW is interacting with the TSV wall. In case of 100 MHz and a 60° acoustic lens the SAW interacts rather close to the bottom of the TSV, whereas for 100 MHz and 80° more towards the open side of the TSV. From time step 4 to 6, the propagation of the direct reflection towards the sample and interference with the SAWs is obvious.

The schematic in Fig. 4c highlights the context between the opening angle of the lens and failure location. The total length of the TSV in Fig. 4c is marked as L1 and L2 and corresponds to the lower and upper half of the TSV height, respectively. Figure 4d depicts two C-scan image data sets with interference fringes utilizing an acoustic lens with a center frequency of 100 MHz and an opening angle of 60° and 80°, respectively. The displayed inhomogeneity within in the interference fringes of the C-scan is more prominently detected for the acoustic lens with 80° than for 60°.

The plot between the normalized intensity of the absolute values of the velocity vectors versus the outer most layer of the TSV height for the same defocus position of 100 MHz 60° and 80° lenses, is shown in Fig. 4e. Figure 4e quantitatively validates the argument that there is a direct dependence between the opening angle of the acoustic lens and defect detectability. That is, the maximum intensity of the acoustic wave generated by the lens with 100 MHz and 60°, as plotted in purple color, is depicted at a height of 102 μm from the bottom of the TSV. Whereas, for 100 MHz and 80° the maximum intensity is observed at a height of 205 μm from TSV bottom, as

plotted in green color. The considerable difference implies that a careful selection of the opening angle from the acoustic lens is crucial for the failure analysis. If the location of the defect is located within the 'L2' region, then the interference fringes for an opening angle greater than 60° show more intensity variations in the SAM C-scan image. This is also highlighted by the intensity-plot across the xy-plane, see Fig. 4d. Figure 4f supports this argument with a SEM image taken from the TSV under investigation. i.e., a crack combined with a micron-sized delamination is observed within the 'L2' length of TSV. The interference fringes of this TSV observed with the 80° lens shows more variations in the C-scan image as well.

## Conclusion

In the realm of failure analysis (FA), the demand for high-resolution information is paramount, but needs to be achieved within the constraints of cost- and time-effectiveness. Here, we tackle this challenge by conducting a SAM interferometry-based approach for TSV failure analysis suitable for in-line inspection. The approach enables the characterization within the industrial environment with high resolution down to the nm-regime. Further, it provides user-friendliness with respect to the sample- to transducer-distance and offers a sufficient penetration depth with up to 250 μm.

The presented approach leverages interference effects between the generated bulk and surface acoustic waves. An accurate defocusing of the transducer as well as the use of specific lenses with an opening angle larger than the associated Rayleigh angle of the material under investigation, makes FA below the nominal resolution limit of a 100 MHz transducer feasible. Indeed, the approach provides evidence that cracks with a crack opening less than about 200 nm are possible, i.e. cracks with an opening less than half a wavelength can be detected with a frequency of 100 MHz within the investigated open TSV geometry. Unlike from the conventional SAM approach that operates with a focused transducer, our methodology utilizes a defocused transducer and a lens with an opening angle larger than the Rayleigh angle. By employing the concept of interferometry, we generate distinct characteristic interference patterns within the C-scan image data. The differentiation of the generated patterns allows the application of automated TSV classification and localization routines based on machine learning[17]. The experimental results, are complemented by EFIT simulations. The latter validate the generation of the SAWs utilizing the SAM interferometry setup as well as provide a deeper understanding about the SAW propagation and impact of the lens angles. We argue that the approach is not only suitable for open TSV technologies. The simulation result in Supplementary Fig. 7 indicates a similar excitation of SAWs for filled TSVs which suggests a utilization of the acoustic interferometry approach beyond open TSV technologies. An extension to smaller TSV geometries, than investigated here is also feasible, however will demand higher frequencies and specifically tailored pulse manipulation.

Additionally, we emphasize based on the experimental and simulation results that for the FA it is essential to consider the opening angle of the acoustic lens. The angle has an unavoidable impact on the sensitivity of the acoustic wave and, consequently, influences the lens selection process. Specifically, our finding implies that the acoustic lenses with angles larger than 60° prove to be more suitable in detecting inhomogeneities towards the open end of the TSV.

In conclusion, the SAM interferometry-based approach pushes the boundaries of modern failure analysis. We point out that the discussed acoustic interference approach is not only limited to failure analysis of TSVs but can be also exploited to further topics.

## Methods

### Scanning Acoustic Microscope (SAM) & Acoustic Interferometry Approach

The SAM setup used in this study is a 'SAM 400' by PVA TePla Analytical Systems GmbH, Westhausen, Germany. We performed the analysis of the reflected signal from the sample (A-scan signal) as well as the cross-sectional image of the sample (C-scan image) using the WinSAM 8.0.2293.0 software developed by PVA TePla with a time resolution of 0.1429 ps using a 7 GHz

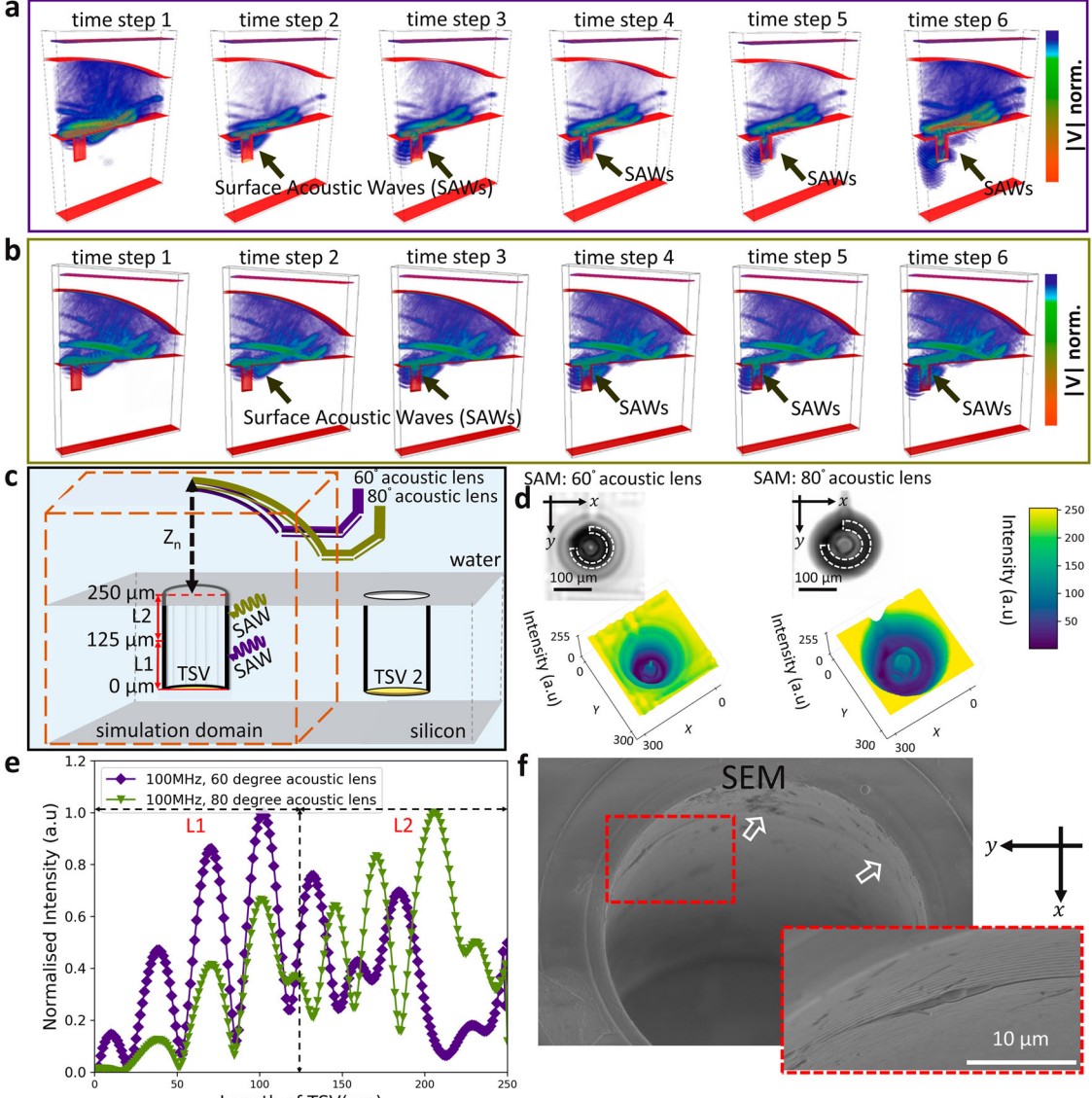

**Fig. 4 | Impact of the opening angle from the lens on failure analysis.** The 3D EFIT simulation shows the interaction of the SAWs with the metallized TSV wall for different time steps. The position of the lens is kept the same for both opening angles. Excited SAWs using a lens with an opening angle of (**a**) 60° and **b** 80°. The point of contact for the SAWs utilizing the lens with 100 MHz, 60° is more in the vicinity of the bottom, whereas for 100 MHz and 80° more towards the open end of the TSV (time steps 1–6). Intensity for the velocity ranges from blue (high) to red (low). **c** Schematic illustration showing the influence of the opening angle of the lens on the SAW interaction with the TSV. **d** SAM C-scan image (above) and intensity representation (from dark purple to yellow) in the x-y-plane of a single TSV obtained using 100 MHz, 60° (purple) and 100 MHz, 80° (green). The observed inhomogeneity within the pattern at $Z_7$ is highlighted (white dashed line) to guide the eye for the two different opening angles. **e** EFIT simulation of the normalized intensity of the absolute values of velocity vectors at the outermost layer of the TSV versus the length of the TSV for 60° (purple) and 80° (green). **f** SEM image measured on the same TSV. A crack running along the sidewall of the TSV as well as a micron-sized delamination is visible. The latter seems to define to the characteristic *Class 2* pattern. See also below the magnified image highlighted with a dashed red frame.

ADC card. To perform acoustic interferometry, we used a 100 MHz transducer affixed to the flat surface of an acoustic lens with a gold coating. The lateral resolution in SAM is related to the wavelength of the acoustic waves($\lambda$) and the numerical aperture (*NA*) by

$$W = \frac{\lambda}{2NA} \qquad (1)$$

For a transducer frequency of 100 MHz, half the wavelength corresponds to approximately 15 μm in water and a penetration depth of 0.4 mm, whereas the axial resolution depends on the longitudinal wave velocity (*c*)

and temporal pulse width of the acoustic wave ($\delta_t$) by Eq. (2).

$$\delta_{axial} = \frac{c}{2}\delta_t \qquad (2)$$

By assuming the speed of sound in water as 1450 meters per second and a temporal pulse width of 10 ns, the axial resolution of the SAM setup is 7.56 μm. By defocusing the acoustic lens with an opening angle greater than the Rayleigh angle, we excited SAWs and detected cracks with opening below the lateral and axial resolution of conventional SAM. SAWs can be efficiently generated even by slightly defocusing the transducer below the focal point, as long as the lens has an opening angle above the critical Rayleigh angle. A key challenge is to select the optimal defocus position that

yields the best contrast in the C-scan image with maximum information about the TSV. Here, the movement of the transducer in z-direction is crucial. The maximum distance from the focal point in z-direction of the transducer is $-240\,\mu m$. As the transducer is brought closer to the sample surface, the C-scan image exhibits reduced contrast, making it challenging to discern the underlying interference pattern. Our analysis shows that a step size of 20 μm along z-direction provides the best possible result. Among all the Z sets, $Z_7$ is found to show the best defocus position, see Supplementary Note 1.

## Tone burst

The tone burst box is incorporated with an arbitrary waveform generator (AWG) that has been programmed with different burst duration. The AWG helps to generate the necessary electrical burst for the excitation of the acoustic waves. The length of the burst is set typically between 20 ns to 100 ns. We select 50 ns duration of the shaped tone burst. The frequency is set in accordance with the bandwidth of the central frequency provided by the transducer with 100 MHz. Here, we use a tone burst frequency of 113 MHz. The desired tone-burst RF electrical signal has been generated using 'tone burst control software' developed by PVA TePla. The trigger for the piezo electric transducer is the intermittent RF pulse from the tone burst box. There are two types of excitation signals namely wideband and narrow band that can be used to generate acoustic signals. The selection of excitation signal depends on the complexity of the sample. Narrow band signals help in providing high spatial resolution images of complex samples like TSV wafers and are capable in detecting small features in the sample.

## Scanning Electron Microscope (SEM)

The field emission SEM is used as complimentary tool to correlate the SAM interferometric pattern of TSVs with conventionally obtained defect data. The SEM setup is equipped with a 360° rotation stage. We utilize the secondary electron (SE) detector to image the TSVs.

## Acoustic lens

For our SAM interferometry approach, we use a custom engineered gold lenses with a spherical cavity of 60° and 80° on a sapphire rod. To mitigate the energy loss from the impedance mismatch between the lens material and water, an anti-reflective coating (ARC) with one-quarter wavelength of the acoustic wave is applied onto the lens opening.

The limited working distance of the 80° lens provides a lower contrast within the C-scan image. Therefore, we augmented the gain of the A-scan signal using WinSAM software to enhance the contrast.

## Samples

ams-OSRAM AG, Premstätten, Austria (AMS) provided the silicon wafers (200 mm) with TSVs containing artificially induced defects at unknown locations. The analyzed open TSV technologies in this study are with a diameter and length of 100 μm and 250 μm, respectively. The TSVs are fabricated following the 'via last' approach. The sidewall of the TSVs consist of various layers – $SiO_2$ as insulation layer, tungsten as conductive layer metal, $SiO_2$ and SiN for passivation. The open TSVs are used over filled (closed) TSV technology, due to obvious advantage with respect to reduced thermomechanical stresses.

## ML-based failure localization and classification

We used two CNN models arranged in an E2E fashion to localize and classify TSVs from the test array[17]. The first CNN is dedicated for localizing all the TSVs from the SAM C-scan image. The input to the second CNN are the thus detected TSVs from CNN1. Here, the TSVs are classified into three classes according to the defect level. All the TSVs with homogeneous fringes are classified as *Class 1* (non-defective TSVs) and all other TSVs with inhomogeneous fringes can be classified according to the underlying failure class e.g. *Class 2, 3*, etc. For further details see ref. 17.

## 2D EFIT

We utilize Python 3.7 and tensor flow version 2.1.0, to implement the necessary equations for the 2D and 3D EFIT simulations. For this study, we simulate a simple model of open TSV technology with layers such as $SiO_2$, tungsten, SiN and layer thicknesses corresponding to real structure of the TSV. The necessary differential equations for EFIT simulations of acoustic wave propagation have been discretised into finite difference equation (see, Supplementary Note 2). We consider the entire simulation domain to be composed of several material grids. Within such a grid, the material is assumed to be constant. The Courant–Friedrichs–Levy (CFL) condition determines the numerical stability for the EFIT simulation[67,73] and is given by

$$\Delta t \leq \Delta t_{max} = \frac{1}{C_l \sqrt{\frac{1}{(\Delta x)^2} + \frac{1}{(\Delta y)^2} + \frac{1}{(\Delta z)^2}}} \qquad (3)$$

where $\Delta t_{max}$ is the maximum possible time step, $C_l$ is the fastest longitudinal wave speed in elastic medium and $\Delta x, \Delta y, \Delta z$ are the lengths of unit cell. For simplicity equal lengths, $\Delta x = \Delta y = \Delta z = \Delta s$ are considered. We assign 10 grid points per shear wavelength in our simulation in order to achieve a balance between computational cost and accuracy of the representation of the dispersion (i.e. the numerical error in phase and group velocity of the wave).

Furthermore, we utilize the Avizo software version 2022.2 to perform the 3D visualization of the geometry and velocities at different time steps.

## 3D EFIT

We develop the 3D-EFIT code for simulating the SAM setup utilizing Python. We reduce the simulation area so as to deal with large sizes of velocity and stress arrays, while performing the simulation in three dimensions. By utilizing the Avizo software, the normalized values of the absolute velocity at different time steps along with the SAM setup and sample are visualized. We deal with the computational power required for solving the nine arrays using the memory mapping from 'numpy' library. This helped in accessing the arrays from the disk rather than reading the entire file into the memory. The parallel computing of data is effectively realized by executing the EFIT script on the GPU.

## Data availability

All data that support the findings of this study are available from the corresponding author upon reasonable request.

## Code availability

All codes that support the findings of this study are available from the corresponding author upon reasonable request.

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

## Acknowledgements

We acknowledge the financial support by Die Österreichische Forschungsgesellschaft (FFG) under Bridge Young Scientist, Proj. No. 872629, "REFORM" and partly under the scope of the COMET program within the K2 Center "Integrated Computational Material, Process and Product Engineering (IC-MPPE) (Proj. No 886385, P2.22). This program is supported by the Austrian federal Ministries for Climate Action, Environment, Energy, Mobility, Innovation and Technology (BMK) and for Labour and Economy (BMAW), represented by the Austrian Research Promotion Agency (FFG), and the federal states of Styria, Upper Austria and Tyrol. We acknowledge the necessary experimental support from E. Kozic (Materials Center Leoben (MCL) Forschung GmbH) and M. Gritzner (ams-OSRAM AG) for SEM measurements.

## Author contributions

R.B. planned the SAM measurements and modifications towards the acoustic interferometry setup. P.P. conducted the SAM measurements and performed the modifications. J.S. provided the TSV wafers for analysis. I.W. supported the modification of the SAM set up with development of tone burst module and custom engineered acoustic lenses. P.P. performed the analysis under supervision of R.B. R.H. generated the original EFIT code. P.P. performed the EFIT simulations and enhanced the code. R.B. and P.P wrote, revised and improved the manuscript. All authors discussed the results and commented on the paper.

## Competing interests

The authors declare no competing interests.
