## [Peer Review File · Communications Engineering]

Reviewers' comments:

Reviewer #1 (Remarks to the Author):

This manuscript presents a non-destructive failure detection approach by applying acoustic interferometry in a SAM-setup, which is validated in the detection of nm-cracks in TSVs. The propagation and interaction of excited SAWs in the TSVs are revealed by comprehensive EFIT simulations. In-depth discussions on the adjustment of the transducer position and the influence of the opening angle of the lens are also carried out. As the cracks in TSVs have serious impacts on the reliability of integrated systems, such a non-destructive and high-resolution failure detection approach is beneficial and significant. However, there are still several issues to be addressed.

Major issues:

1. As mentioned by the authors, there are multiple failure detection methods including some non-destructive ones like X-CT which can provide the quality assessment of hundreds of TSVs. A more comprehensive comparison between the proposed approach and state-of-the-art techniques would be helpful for highlighting the novelty of this work.

2. The dimension and structure of the measured TSV is fixed in this work. Are they selected according to a specific application? How is the feasibility of the proposed approach for TSVs with other dimensions or structures? For example, the central area of such deep TSVs with annular conductor is usually filled by materials like polymers for avoiding oxidation and better stability. However, a central filler will affect the utilization of the proposed approach. Are there any solutions or does this failure detection have to be conducted before the filling step? Moreover, this approach is also not suitable for TSVs with fully filled conductors including the high-aspect-ratio TSVs that are favorable for the miniaturized integrated systems. Therefore, it is suggested to indicate the feasible TSV structure or application domains of the proposed approach.

3. In addition, can the authors provide discussion on the correlation between the key setup parameters of the proposed approach and the TSV dimensions? In other words, what parameters need to be adjusted according to the TSV dimensions. Besides, does the pitch between adjacent TSVs influence the detection results?

4. According to the authors, the interference fringes for 'Class 1' and 'Class 2' are similar to reported works without involving the optimized acoustic interferometry technique in this work while

the one for 'Class 3' is newly observed. Can the authors present the results from conventional SAM-setup without the novel technique in this work for better comparison?

Minor issues:

5. Even though that the end-to-end convolutional neural network has been reported in ref. 17 as mentioned by the authors, it is suggested to add some concise descriptions about this data curing procedure for better integrity.

6. There is a mistake on the coordinate values of the Horizontal Axis of Fig. S1c.

7. Is the SEM image in Fig. 2f from 'TSV(C1a)'? The crack seems to be different from the one in Fig. 2e.

8. As the opening angle of the lens has an impact on the detection possibilities, should multiple opening angles be combined in a single detection procedure for more comprehensive detection of the whole TSV sidewall or just an angle is enough? Please indicate the opening angle of the lens used in the 'Acoustic Interferometry for high-resolution sub-micron crack detection' section.

9. The manuscript is well-written. However, some typos need to be corrected. For example, the 'defocussed' and 'defocussing' should be 'defocused' and 'defocusing', and the 'utilising' in Line 484 should be 'utilizing'. And one apostrophe in Table 1 is wrong.

Reviewer #2 (Remarks to the Author):

The manuscript presents the research work on improving the capability of scanning acoustic microscopy (SAM) for inspection of 3D IC packages. The proposed concept addresses the critical need for SAM to be capable of high-accuracy inspection. I believe that the technology has lots of potential. That being said, there are several issues that need to be addressed before the paper can be accepted.

1. The authors need to improve their literature survey. For instance, there are review papers on nondestructive volumetric inspection techniques (e.g., SAM, XRM, etc.) for 3D packaging, and SAM

technology by itself. Those publications are believed to provide more insight for the overall background.

2. The balance of penetration and resolution needs to be elaborated. Are they competing objectives in your method?

3. Key information on simulation needs to be provided. For instance, are there any governing equations? A flowchart on simulation processes would be helpful.

4. Also, it is not clear to me why simulation is needed?

5. The results do not seem to demonstrate the sub-micron capability clearly. This needs to be strengthened in the revision.

Point-by-Point

Dear Reviewers,

Thank you for the careful reading of the manuscript. Please see below our response to your questions and comments.

Reviewers comments:

Reviewer #1 (Remarks to the Author):

This manuscript presents a non-destructive failure detection approach by applying acoustic interferometry in a SAM-setup, which is validated in the detection of nm-cracks in TSVs. The propagation and interaction of excited SAWs in the TSVs are revealed by comprehensive EFIT simulations. In-depth discussions on the adjustment of the transducer position and the influence of the opening angle of the lens are also carried out. As the cracks in TSVs have serious impacts on the reliability of integrated systems, such a non-destructive and high-resolution failure detection approach is beneficial and significant. However, there are still several issues to be addressed.

We thank reviewer for the careful reading and valuable input to enhance the quality of the manuscript. Please see all the answers to the addressed questions below.

Major issues:

1. As mentioned by the authors, there are multiple failure detection methods including some non-destructive ones like X-CT which can provide the quality assessment of hundreds of TSVs. A more comprehensive comparison between the proposed approach and state-of-the-art techniques would be helpful for highlighting the novelty of this work.

Thank you for your thoughtful comment. We addressed the comment within the manuscript and revised the introduction of the manuscript accordingly. In particular we provided a more comprehensive insight about μ -XCT and XRM possibilities within the introduction, as well as highlighted the novelty of this work more in the result section "acoustic interferometric setup". We also added here more references in context to recent studies of microelectronic devices.

2. The dimension and structure of the measured TSV is fixed in this work. Are they selected according to a specific application? How is the feasibility of the proposed approach for TSVs with other dimensions or structures? For example, the central area of such deep TSVs with annular conductor is usually filled by materials like polymers for avoiding oxidation and better stability. However, a central filler will affect the utilization of the proposed approach. Are there any solutions or does this failure detection have to be conducted before the filling step? Moreover, this approach is also not suitable for TSVs with fully filled conductors including the high-aspect-ratio TSVs that are favorable for the miniaturized integrated systems. Therefore, it is suggested to indicate the feasible TSV structure or application domains of the proposed approach.

Thank you for your question. In this work, we analyse an open TSVs technology with a dimension of 100 μ m in diameter and 250 μ m in height. We added in Fig. 1 the geometry of the structure. The paper highlights that here with 100 MHz, cracks with an opening of about 200 nm can be detected. The detected feature size is significantly smaller than the lateral resolution depicted from the frequency of the transducer. Since there is no state-of-the-art characterization method in particular for open TSV technology with respect to sub-micron resolution, time and statistical yield as well as cost efficiency,

Point-by-Point

an interferometric ultrasound-based characterization approach has been developed, as introduced within the submitted manuscript.

The proposed SAM interferometry approach has been developed to analyse the quality of the TSVs after the metallization step in the production line. Usually only unwanted residuals are left within the TSV after this stage. The method is suitable to detect those residuals as well as cracks originating from generated stress within the metallized side wall.

In context to your question regarding TSVs filled with a polymer we performed simulations to study in addition the excitation of surface acoustic waves for such TSVs. Please see the modifications in this context in the supplementary Note 4 as well as changes in the manuscript, see result section "Excitation of surface acoustic waves within the TSV" as well as in the conclusion. The simulation confirms the excitation of SAWS and exhibits a similar behaviour seen for the non-filled TSV. Once surface acoustic waves can be excited, an interaction with the TSV structure is possible. The simulation result in Supplementary Fig. S7 indicates a similar excitation of SAWs for filled TSVs and therefore suggest an extension of the interferometry approach beyond open TSV technologies.

We argue that the interferometric approach offers versatility in defect detection across TSVs with various dimensions. However, a frequency is necessary to resolve the TSV structure with sufficient contrast and resolution. For copper filled TSVs in general dimensions of about 10 μ m in diameter and 100 μ m in depth are used. Therefore, in order to resolve the TSVs, transducers with frequencies larger than 100 MHz are necessary.

In addition, the pulse manipulation is key. We added in the result chapter "acoustic interferometry setup" further information as well as in the conclusion a brief discussion in this context.

3. In addition, can the authors provide discussion on the correlation between the key setup parameters of the proposed approach and the TSV dimensions? In other words, what parameters need to be adjusted according to the TSV dimensions. Besides, does the pitch between adjacent TSVs influence the detection results?

Thank you for this question. We have modified the result section "acoustic interferometry setup" accordingly also in context to question 2. We provide more insights about the key parameters also in context to the TSV dimension. Main parameters are the frequency as well the pulse duration. An essential element in this context displays the tone burst setup.

In this study, the pitch length between the adjacent TSVs is approximately 350 μ m. From the simulation results presented in supplementary Fig S9, it has been observed that the excitation and interaction of SAWs depends on the position of the acoustic lens with respect to the TSV. This observation is also detailed in the revised manuscript. Please see the modification in Supplementary Note 5 as well in the result section "Excitation of surface acoustic waves within the TSV".

4. According to the authors, the interference fringes for 'Class 1' and 'Class 2' are similar to reported works without involving the optimized acoustic interferometry technique in this work while the one for 'Class 3' is newly observed. Can the authors present the results from conventional SAM-setup without the novel technique in this work for better comparison?

We thank the reviewer for this question. We have added the results of C-scan image of TSVs obtained from the conventional SAM setup, employing a 100MHz, 60° acoustic lens as Supplementary Fig. S2. Additionally, conventional and SAM interferometric C-scan images of the same TSVs are acquired with the same frequency of 100 MHz. The C-scan image obtained from conventional SAM does not give information about the quality of the TSVs. Whereas, the C-scan image of TSV obtained using the acoustic interferometry shows the characteristic interference patterns associated with 'Class 1' and

'Class 3', respectively. Those patterns have been thoroughly validated by using SEM microscope. In particular 'Class 3' could be associated to cracks located within the metallization of the TSV. Below the Figure R1 is now also shown as Supplementary Fig. S2.

Figure R1. Comparison of SAM C-scan images of TSVs. The C-scan images with 100MHz of the identical TSVs (TSV 1 and TSV 2) are obtained using (a) Conventional SAM setup and (b) SAM interferometric setup. The TSVs showing interference for the interferometric approach with and without inhomogeneity are represented in orange and green color respectively. The conventional SAM C-scan image does not show any significant difference between the two TSVs.

Minor issues:

5. Even though that the end-to-end convolutional neural network has been reported in ref. 17 as mentioned by the authors, it is suggested to add some concise descriptions about this data curing procedure for better integrity.

Thank you for this valuable input. We have revised the manuscript in this context and added more concise description about the data curing procedure in the result section "acoustic interferometry for high-resolution sub-micron crack detection".

6. There is a mistake on the coordinate values of the Horizontal Axis of Fig. S1c.

Thank you for pointing this mistake. We have checked and corrected the horizontal axis of Fig. S1c.

7. Is the SEM image in Fig. 2f from 'TSV(C1a)'? The crack seems to be different from the one in Fig. 2e.

We thank the reviewer for bringing up this query. This was an unintentional mistake. We have rectified this error and updated the SEM image in Fig. 2f.

8. As the opening angle of the lens has an impact on the detection possibilities, should multiple opening angles be combined in a single detection procedure for more comprehensive detection of the whole TSV sidewall or just an angle is enough? Please indicate the opening angle of the lens used in the 'Acoustic Interferometry for high-resolution sub-micron crack detection' section.

Point-by-Point

Thank you for this question. It seems this was not clearly stated in the manuscript. We revised the manuscript accordingly. We appreciate the reviewer's valuable suggestion, and we have now included the opening angle of the acoustic lens in the 'Acoustic Interferometry for high-resolution sub-micron crack detection' section.

As mentioned, the opening angle of the acoustic lens significantly influences the detection capabilities for TSV defects. The interaction of SAWs with the TSV height varies based on the lens-opening angle, influencing the overall defect detection. While both 60° and 80° lenses can identify defects within the TSV, there observed distinct variations in the intensity within the C-scan interference patterns. Therefore, modifying the lens design itself holds the potential to enhance the comprehensiveness of TSV sidewall defect detection with better resolution.

9. The manuscript is well-written. However, some typos need to be corrected. For example, the 'defocussed' and 'defocussing' should be 'defocused' and 'defocusing', and the 'utilising' in Line 484 should be 'utilizing'. And one apostrophe in Table 1 is wrong.

We thank the reviewer for the careful reading. We corrected the typos as well as the wrong apostrophe in Table 1.

Reviewer #2 (Remarks to the Author):

The manuscript presents the research work on improving the capability of scanning acoustic microscopy (SAM) for inspection of 3D IC packages. The proposed concept addresses the critical need for SAM to be capable of high-accuracy inspection. I believe that the technology has lots of potential. That being said, there are several issues that need to be addressed before the paper can be accepted.

We thank the reviewer for the valuable input and careful reading. Please see the response to all your comments below.

1. The authors need to improve their literature survey. For instance, there are review papers on nondestructive volumetric inspection techniques (e.g., SAM, XRM, etc.) for 3D packaging, and SAM technology by itself. Those publications are believed to provide more insight for the overall background.

Thank you for this feedback. We have carefully reviewed the manuscript and incorporated your suggestions. We have also included discussions on non-destructive volumetric inspection techniques. The following literatures have been added in the introduction to provide a more comprehensive background for our work e.g. :

27. Rau, H. & Wu, C.-H. Automatic optical inspection for detecting defects on printed circuit board inner layers. *Int. J. Adv. Manuf. Technol.* 25, 940–946 (2005).
28. Zhou, W., Apkarian, R., Wang, Z. L. & Joy, D. Fundamentals of scanning electron microscopy (SEM). in *Scanning microscopy for nanotechnology* 1–40 (Springer, 2006).
29. Hsu, P.-N. *et al.* Artificial intelligence deep learning for 3D IC reliability prediction. *Sci. Rep.* 12, 1–7 (2022).
30. Cui, C. *et al.* Correlative, ML based and non destructive 3D analysis of intergranular fatigue

Point-by-Point

- cracking in SAC305 Bi solder balls. *npj Mater. Degrad* (accepted).
31. Aryan, P., Sampath, S. & Sohn, H. An overview of non-destructive testing methods for integrated circuit packaging inspection. *Sensors* 18, 1981 (2018).
 32. Su, Y. *et al.* Volumetric nondestructive metrology for 3D semiconductor packaging: A review. *Measurement* 114065 (2023).
 33. Hartfield, C., Schmidt, C., Gu, A. & Kelly, S. T. From PCB to BEOL: 3D X-ray microscopy for advanced semiconductor packaging. in *2018 IEEE International Symposium on the Physical and Failure Analysis of Integrated Circuits (IPFA)* 1–7 (2018).
 34. Modgil, D. *et al.* Material identification in x-ray microscopy and micro CT using multi-layer, multi-color scintillation detectors. *Phys. Med. & Biol.* 60, 8025 (2015).
 35. Crewe, A. V, Wall, J. & Welter, L. M. A high-resolution scanning transmission electron microscope. *J. Appl. Phys.* 39, 5861–5868 (1968).
 37. Wickramasinghe, H. K. Scanning acoustic microscopy: a review. *J. Microsc.* 129, 63–73 (1983).

2. The balance of penetration and resolution needs to be elaborated. Are they competing objectives in your method?

Thank you for the question. We modified the manuscript in this context, please see the results section “acoustic interferometry setup” as well added a sentence in the result section “acoustic interferometry for high-resolution sub-micron crack detection”.

The balance of penetration depth and the resolution are the key objectives in the proposed method.

The resolution W of an acoustic microscope depends on the frequency of the transducer via $W = \lambda/2NA$. Here, the wavelength is denoted $\lambda = v/f$, and the numerical aperture $NA = \sin(\theta_0)$, where θ_0 is the semi-angle of the lens aperture and v denotes the velocity of the acoustic wave in the propagation medium.

Increasing the frequency of the acoustic wave results in higher resolution, thus smaller structures can be resolved. However, also the attenuation in the coupling liquid and in the sample increases with frequency. For this reason, higher frequency corresponds to lower penetration depth and in many cases, a trade-off between resolution and penetration depths has to be made in the SAM measurements. While higher-frequency waves can enhance resolution, they also lead to increased attenuation in the coupling liquid and sample, which in turn reduces penetration depth. The low frequency acoustic waves helps in achieving greater penetration depth, but also results in a lower resolution or poorer detection limit. Therefore, SAM interferometry approach involves a strategic choice in balancing the penetration depth and resolution. The goal is to utilize a frequency sufficient to resolve the geometry of the structure, like here the TSV. However the generated SAWs are then utilized to resolve features like cracks with significant smaller features sizes than the frequency of the transducer.

3. Key information on simulation needs to be provided. For instance, are there any governing equations? A flowchart on simulation processes would be helpful.

Thank you for the input please see the modifications according to your comment in Fig.3a as well as Supplementary Note 2.

Point-by-Point

The primary governing equations in the 3D EFIT simulation is Cauchy equation of motion and Hooke's law.

$$\rho \frac{\partial v}{\partial t} = \nabla \cdot \sigma + f \quad (1)$$

$$\sigma = \lambda \epsilon \delta + 2\mu \epsilon \quad (2)$$

In the manuscript, Fig. 3a now shows the flowchart illustrating the key steps and processes involved in the 3D EFIT simulation. The simulation workflow is outlined through four key steps: initialization, solver definition, time stepping, and post-processing. The initial stage encompasses the setup of the simulation domain, including the definition of geometries such as the acoustic lens, transducer, and sample, along with their corresponding material parameters. The transducer's excitation is modelled as a modulated sinusoidal wave with a Gaussian envelope, emulating a tone burst signal. The frequency and length of this tone burst signal is set in accordance with the SAM interferometry setup. The input stage also involves determining spatial and time step sizes. The spatial step size is determined by the minimum wavelength of shear waves in the medium. For our simulations, 10 grid points per wavelength is used as recommended in literatures. Additionally, the Courant-Friedrichs-Levy (CFL) condition is applied to calculate the upper limit of the time step size. An absorbing boundary condition is also defined at this stage to attenuate the reflections at the domain boundaries. The second step of the simulation focuses on grid generation based on the calculated spatial step size, followed by defining necessary equations of EFIT. Finite difference equations for stress and particle velocity are derived from integral forms of governing equations (1) and (2). As the simulation progresses in time during the third step, the velocity fields are updated, followed by the updating of stress tensors using the previously updated velocity components. The final stage of the simulation involves post-processing step, as illustrated in the flow chart.

4. Also, it is not clear to me why simulation is needed?

Thank you for the question. It seems the manuscript was not clear in this context. We modified the manuscript accordingly, in particular the summary in the introduction, the result section "Excitation of surface acoustic waves within the TSV" and conclusion.

The primary aim of conducting the EFIT simulation is to verify the excitation of SAWs within the SAM interferometry setup, distinguishing it from conventional SAM methods. Moreover, the EFIT simulation has been carried out to gain a deeper understanding of the excitation, propagation, and interaction of SAWs, with specific focus on the opening angles of acoustic lenses. The results from EFIT simulation thus helps in the optimization of acoustic lens and furthermore to develop a more sophisticated lens.

5. The results do not seem to demonstrate the sub-micron capability clearly. This needs to be strengthened in the revision.

Thank you for the comment. The manuscript was not formulated clearly. We modified the manuscript accordingly especially we extended the result section "Acoustic Interferometry for high-resolution sub-micron crack detection" in this context. Please see the revised paragraph.

REVIEWERS' COMMENTS:

Reviewer #1 (Remarks to the Author):

I think the issues have been well addressed and the paper can be accepted for publication.